# Free-Grazing versus Enclosure Lead to an Increase in the Germination of the *Leymus chinensis* Seed Bank in the Hulunbuir Grassland

**Hongmei Liu [1,†], Yanling Wu [2,†], Yingxin Li [3], Shijie Lv [4], Zhijun Wei [4], Baorui Chen [3], Lijun Xu [3], Guixia Yang [3], Xiaoping Xin [3,\*] and Ruirui Yan [3,\*]**

1. Grassland Institute of Inner Mongolia Academy of Forestry Sciences, Hohhot 010010, China
2. College of Life Science and Technology, Inner Mongolia Normal University, Hohhot 010122, China
3. Institute of Agricultural Resources and Regional Planning, Chinese Academy of Agricultural Sciences, Zhonguancun South Street, Haidian District, Beijing 100081, China
4. College of Science, Inner Mongolia Agricultural University, Hohhot 010018, China
* Correspondence: yanruirui@caas.cn (R.Y.); xinxiaoping@caas.cn (X.X.); Tel.: +86-10-8210-9741 (R.Y.)
† These authors contributed equally to this work.

**Abstract:** *Leymus chinensis* is a primary plant in the meadow steppe and typical steppe of China. With global warming and increasing grazing intensity, grassland degradation is being exacerbated. To better protect the *L. chinensis* grassland in this area and provide a theoretical basis for restoring it, this paper compared the germinable seed bank in the soil and the germination characteristics of *L. chinensis* (including initial germination time, duration of germination, germination termination time, germination dynamics, and germination index) in free-grazing and enclosed areas. At the same time, combining information about the density of *L. chinensis* on the ground and previous research results, a comprehensive analysis was conducted. The major results were: (1) there was no significant difference in the number of germinable seeds in the soil between the free-grazing area and the enclosed area, and these seeds were mainly concentrated in the 0–2 cm soil layer. (2) The free-grazing area resulted in a significant increase in the number of germinable *L. chinensis* seeds and advanced the initial germination time. (3) The number of soil germinated seeds and the number of *L. chinensis* germinated seeds decreased with the increase in soil depth. (4) Livestock grazing behaviours increased seed burial, thus improving the *L. chinensis* germination rate. At the same time, the *L. chinensis* seed bearing percentage and seed quality and the number of germinable *L. chinensis* seeds were significantly higher in the free-grazing area than in the enclosed area. However, this result still needs to be further explored.

**Keywords:** *Leymus chinensis*; soil seed bank; germination rate; sexual reproduction; asexual reproduction

## 1. Introduction

Grazing, through livestock foraging and trampling behaviours, can affect grassland plant species composition and abundance [1–3], leading to changes in species richness or abundance in grazing and nongrazing areas and impacting the species richness or abundance of the soil seed bank [4]. A soil seed bank refers to the sum of active, dormant, and nondormant seeds in a specific soil volume [5]. Soil seed banks provide diversity and are a driving force for all plant communities, and they are also critical to fully understanding vegetation dynamics and mechanisms, maintaining populations [6,7], and restoring natural vegetation [8]. At the same time, a soil seed bank provides the necessary material basis for restoring and reconstructing a grassland after large-scale and long-term disturbances [9]. The species composition and abundance of grassland plant communities are affected by soil seed banks [10] and above-ground vegetation [11].

Grasslands are the largest terrestrial ecosystem in China, and they cover more than 60% of the global land area and account for a quarter of China's total land area [12,13].

The Hulunbuir grassland in Northeast China is a significant green ecological barrier in the north. *Leymus chinensis* is a constructive species of the meadow grassland and plays a leading role in the structure and function of its plant communities. At the same time, due to its absolute predominance in the species abundance of plant communities, it should have an absolute predominance in the soil seed bank regardless of grazing or non-grazing areas. Therefore, research on the seed germination characteristics and seed banks of the *L. chinensis* population has attracted extensive attention [14–16]. Related studies suggest that the species composition and abundance of the soil seed bank are directly affected by aboveground communities [17–20]. However, some studies also showed that species with high seed yields typically dominate the seed bank, but these species may not be dominant in the vegetation [21,22]. Gonzalez [23] also found that the similarity between the species composition size of a grassland seed bank and grassland vegetation cover is generally low. Therefore, the difference between the soil seed banks in a *L. chinensis* meadow grassland between long-term free grazing and no grazing needs to be comparatively studied.

To address the ecological problems caused by continuous grazing, the Chinese Central Government has implemented policies that ban grazing and moderate grazing to promote grassland ecological restoration [24]. It is generally believed that moderate grazing can promote grassland plant growth, maintain biodiversity and productivity, and regulate the dynamics and spatial distribution of dominant populations of plant communities [25–27]. A corresponding grazing prohibition study found that long-term grazing prohibition was unfavourable to dominant plant community populations, species diversity, and plant community productivity functions [5]. Therefore, more empirical research is needed to determine whether the germination seed bank in the soil is consistent with the constructive species under the conditions of long-term moderate grazing and long-term nongrazing (grazing prohibition). At the same time, the seeds of *L. chinensis* have physiological dormancy characteristics [28]; thus, whether the germination characteristics of the soil seed bank are consistent among different grasslands also needs to be studied.

Therefore, based on the meadow steppe with *L. chinensis* as the dominant species, by studying the effect of free-grazing grassland areas and enclosed grassland areas over a long period (since 2006) on the number of germinable seed banks in the soil, the number of germinable *L. chinensis* seed banks in the soil was determined and the *L. chinensis* seed germination characteristics were compared and analysed. The main purpose is to address the following two questions: (1) Are the constructive species *L. chinensis* and plant communities affected by grazing, and are the change rules of germinable soil seed banks consistent? (2) Can grazing affect the seed germination characteristics of the soil *L. chinensis* seed bank? Addressing these relevant issues can provide the necessary theoretical and data support for sustainable grazing and use of the *L. chinensis* and vegetation reconstruction of this degraded meadow grassland.

## 2. Materials and Methods

### 2.1. Overview of the Experimental Site

This study was conducted in a long-term observation sample site in the *L. chinensis* meadow grassland at the Hulunbeier Grassland Ecosystem Experimental Station, Chinese Academy of Agricultural Sciences. The sampling sites were located at Team 6 (*L. chinensis* + forb meadow steppe), Team 11 (*Stipa baicalensis* meadow steppe) and Team 12 (*L. chinensis* meadow steppe) of the Xiertala breeding cattle farm, Hulunbeier city, Inner Mongolia.

### 2.2. Soil Seed Bank Sampling

Nine 1 m × 1 m quadrats were randomly selected in the free-grazing area and the enclosed areas at Teams 6, 11, and 12 on the cattle breeding farm in early October in 2019. Six sampling points in each quadrat were randomly selected by a soil auger with a diameter of 5 cm, and stratified sampling was performed at 0–2 cm, 2–5 cm, and 5–10 cm. The soil of the same soil layer and the same quadrat was then mixed as a soil sample of the

corresponding soil layer of the quadrat, with an area of $6 \times 2.5 \times 2.5 \times 3.14 = 117.75$ cm$^2$. The soil sample volumes of the 0–2 cm, 2–5 cm, and 5–10 cm soil layers of each sample were 235.50 cm$^3$, 353.25 cm$^3$, and 588.75 cm$^3$, respectively. Each soil sample was packaged in a bag and brought back to the laboratory for air drying. Therefore, the sample size was 3 sites $\times$ 2 grassland use areas $\times$ 9 quadrats $\times$ 3 layers of soil samples = 162.

### 2.3. Seed Bank Germination Test

A germination plate (17 cm in diameter and 3 cm in height) was used. Each soil sample on the germination plate had a thickness of 1–2 cm, vermiculite was used as the matrix, and germination occurred for 45 days (note: the soil sample here refers to the original soil sample that was dried with stones and litter removed, Figure 1). Water was sprayed irregularly during the period to ensure that the soil sample was wet. During germination, the species of sprouting seedlings were identified using the seedling and seed morphological characteristics methods, combined with seedling colour, seedling odour, and seed germination characteristics. For the species that could be identified, the number of seed germinations was recorded daily. For the species that could not be identified, in addition to the number of seed germinations recorded daily, they were transplanted for later species identification.

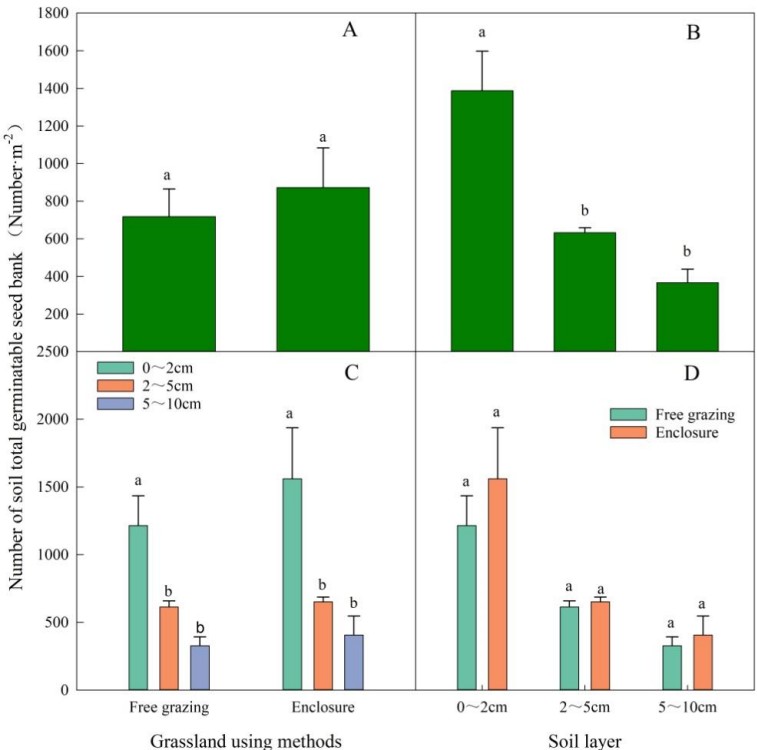

**Figure 1.** Effects of grassland use methods and soil layer on soil total germinable seed bank. Note: Lowercase letters represent significant differences between different using methods or different soil layers ($p < 0.05$). Subfigure (**A**) shows number of soil total germinatable seed bank at 0–10 cm under different using methods. Subfigure (**B**) shows number of soil total germinatable seed bank of free grazing and enclosure at different soil layers. Subfigures (**C**,**D**) show number of soil total germinatable seed bank under different using methods and soil layers.

### 2.4. Data Analysis

Data test: The normality test was carried out on the number of germinated seeds in the soil per m$^2$ (number·m$^{-2}$), the number of germinated *L. chinensis* seeds per m$^2$ (number·m$^{-2}$), the initial germination time, the continuous germination time, the germination termination time, and the germination index of *L. chinensis*. The SAS statistical software UNIVARIATE process was used, and the NORMAL keyword was selected. When

the data did not conform to the normal distribution, square or anti-sinusoidal transformation was used.

Statistical analysis: Variance analysis was performed on the number of germinated seeds in the soil, the number of germinated *L. chinensis* seeds, the number of germinated *L. chinensis* seeds accounting for the number of germinated seeds in the soil, the initial germination time, the continuous germination time, the germination termination time, and the germination index of *L. chinensis*. First, two-factor ANOVA of the soil layer and free-grazing area and enclosed area was conducted. Then, ANOVA of the free-grazing and enclosed areas for the same soil layer and different soil layers in the free-grazing and enclosed areas was conducted. Finally, Duncan's multiple comparison test was conducted for the indicators with significant results from the variance analysis (the Levene's variance homogeneity test was conducted simultaneously in the single factor ANOVA). SAS software was used for ANOVA and normality test. The results of the comparative analysis of the number of germinated seeds in soil, the number of germinated *L. chinensis* seeds, and the number of germinated *L. chinensis* seeds accounting for the number of germinated seeds in soil were plotted into a histogram in SigmaPlot 14.0, and the error line was marked. At the same time, the multiple comparison results were characterized by the letter labelling method (note: there was no significant difference between the same letters representing the comparison, $p > 0.05$). The comparison results of the initial germination time, continuous germination time, germination termination time, and germination index of *L. chinensis* were drawn in a table (note that the data in the table are the mean $\pm$ SE).

Germination dynamics of *L. chinensis*:

- *L. chinensis* germination data were collected over 45 days. Additionally, the numbers of germinable *L. chinensis* seeds in the enclosed and free-grazing areas in the 0–2 cm, 2–5 cm, and 5–10 cm soil layers were compared.
- The data were summarized and averaged.
- A curve chart was drawn in SigmaPlot 14.0 to compare the germination characteristics of *L. chinensis* in the 0–2 cm, 2–5 cm, 5–10 cm, and 0–10 cm soil layers in the free-grazing areas and enclosed areas.

$$GI = \sum_{t=1}^{n} G_t / D_t$$

where $G_t$ is the number of *L. chinensis* seeds germinated every day, and $D_t$ represents germination days, $t$: 1 d, 2 d, . . . , 45 d.

## 3. Results

### 3.1. Effects of the Grazing Area and Enclosed Area on the Soil Germinable Seed Bank

Although the soil germinable seed bank in the enclosed area was slightly larger than that in the free-grazing area, there was no significant difference ($p < 0.05$, Figure 1A). Among the different soil layers, the number of germinable seeds in the 0–2 cm soil layer was significantly higher than that in the 2–5 cm and 5–10 cm soil layers (Figure 1B). In the enclosed area and free-grazing area, the number of soil germinable seeds in the 0–2 cm soil layer was significantly higher than that in the 2–5 cm and 5–10 cm soil layers (Figure 1C), and the number of germinable seeds in each soil layer in the enclosed area was slightly higher than that in the free-grazing area. However, the ANOVA results showed that there was no significant difference in the number of germinable seeds in the soil between the enclosed area and the free-grazing area ($p > 0.05$, Figure 1D). The free-grazing area and enclosed area generally had no significant impact on grassland soil-germinable seed banks. With increasing soil depth, the soil-germinable seed bank showed a downwards trend, and many germinable seeds were stored in the 0–2 cm soil layer.

### 3.2. Effects of Grazing Area and Enclosed Area on the Germinable Seed Bank of Leymus chinensis in Soil

In the soil seed bank, the number of germinable *L. chinensis* seeds in the free-grazing area was significantly higher than that in the enclosed area (Figure 2A). Among the different soil layers, the number of germinable *L. chinensis* seeds is consistent with the change rule of the soil germinable seed bank, and was considerably higher in the 0–2 cm soil layer than in the 2–5 cm and 5–10 cm soil layers (Figure 2B). In the free-grazing area, the number of germinable *L. chinensis* seeds in the 0–2 cm soil layer was significantly higher than that in the 2–5 cm and 5–10 cm soil layers. In the enclosed area, although this variation existed, the ANOVA results were not significant (Figure 2C). The difference in the number of germinable *L. chinensis* seeds between grazed and ungrazed in each soil layer was not significant. However, in all soil layers, the number of germinable seeds in the free-grazing area was significantly higher than that in the enclosed area (Figure 2D). The free-grazing areas promoted the germination of the germinable *L. chinensis* seed bank, and the number of germinable *L. chinensis* seed banks in the 0–2 cm soil layer was relatively higher than that in the other layers.

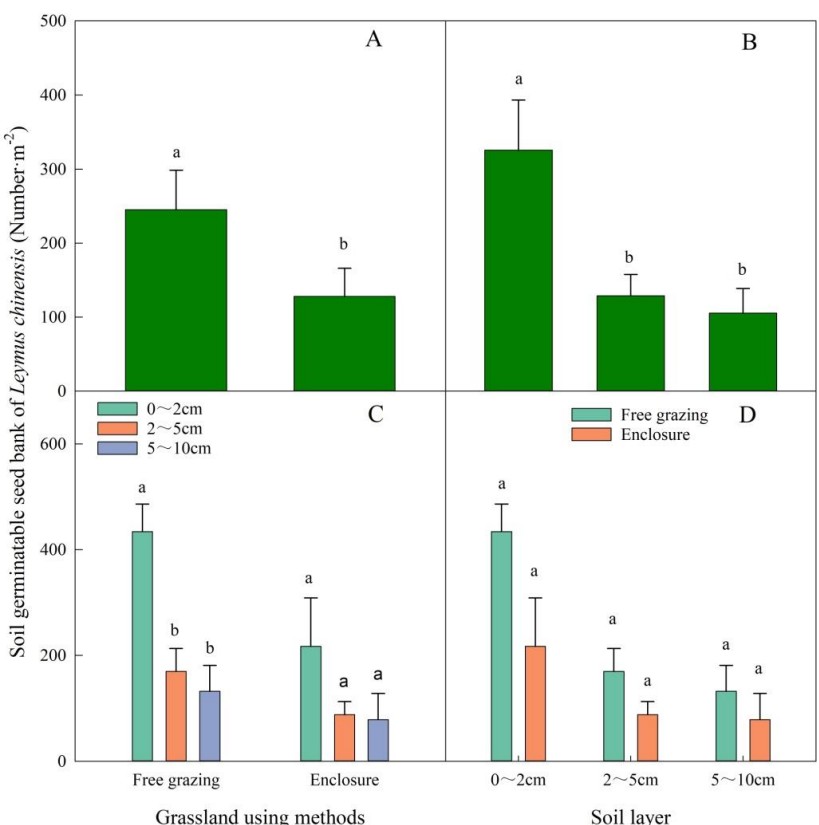

**Figure 2.** Effects of grassland use and soil layer on the soil germinable *Leymus chinensis* seed bank. Note: Lowercase letters represent significant differences between different using methods or different soil layers (*p* < 0.05). Subfigure (**A**) shows total soil germinatable seed bank of *L. chinensis* at 0–10 cm under different using methods. Subfigure (**B**) shows total soil germinatable seed bank of *L. chinensis* of free grazing and enclosure at different soil layers. Subfigures (**C**,**D**) show soil germinatable seed bank of *L. chinensis* under different using methods and soil layers.

### 3.3. Effects of Grazing and Enclosed Areas on the Ratio of the Germinable Leymus chinensis Seed Bank to Overall Soil Germinable Seed Bank

The ratio of the number of germinated *L. chinensis* seeds to the total number of soil-germinated seeds showed that the number of germinated *L. chinensis* seeds in the free-grazing area was significantly higher than that in the enclosed area. In addition, as shown in Figure 3A, the proportion of germinated *L. chinensis* seeds in the number of soil germinated

seeds was significantly higher in the free-grazing area than in the enclosed area. Combining Figures 1 and 2, due to the high number of germinated seeds in the enclosed area (no difference compared with that in the free-grazing area) and the low number of germinated *L. chinensis* seeds in the enclosed area (significant difference compared with that in the free-grazing area), it can be inferred that the free-grazing area led to an increase in the proportion of *L. chinensis* in the plant community, and the number of germinable seeds and its ratio to the total number of soil germinable seeds increased. Between the different soil layers, there was no significant difference in the ratio of the number of germinable *L. chinensis* seeds to the total number of soil germinable seeds (Figure 3B). Similarly, between the enclosed area and free-grazing area, there was no significant difference in the ratio of the number of germinable *L. chinensis* seeds to the total number of soil germinable seeds (Figure 3B). However, notably, the proportion of soil germinated seeds in the enclosed area has an increasing trend with increasing soil depth (Figure 3D). Comparing the proportion of germinated *L. chinensis* seeds in the 0–2 cm, 2–5 cm, and 5–10 cm soil layers in the enclosed area with that in the free-grazing area, the difference in each soil layer was not significant, but the ratio of germinated *L. chinensis* seeds in each soil layer in the free-grazing area was higher than that in the enclosed area.

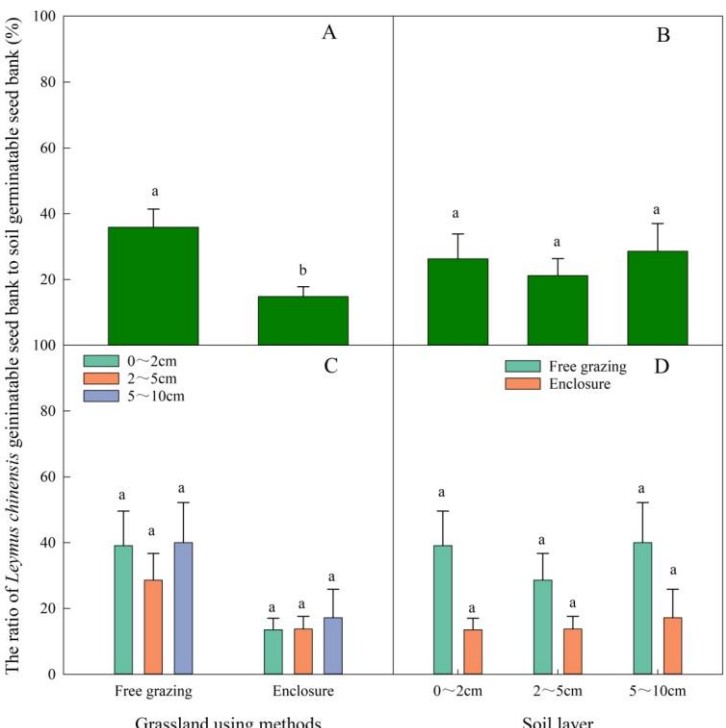

**Figure 3.** Effects of grassland use and soil layer on the ratio of germinable *Leymus chinensis* seed bank to soil germinable seed bank. Note: Lowercase letters represent significant differences between different using methods or different soil layers ($p < 0.05$). Subfigure (**A**) shows the ratio of total germinated *L. chinensis* seeds at 0–10 cm to the total number of soil germinated seeds at 0–10 cm under different using methods. Subfigure (**B**) shows the ratio of total germinated *L. chinensis* seeds of free grazing and enclosure to the total number of soil germinated seeds at different soil layers. Subfigures (**C,D**) show the ratio of germinated *L. chinensis* seeds to the number of soil germinated seeds under different using methods and soil layers.

### 3.4. Effects of the Grazing and Enclosed Areas on the Germination Dynamics of the Leymus chinensis Seed Bank

The number of germinable seeds in the germinable *L. chinensis* seed bank was higher in the free-grazing area than in the enclosed area in the 0–2 cm, 2–5 cm, 5–10 cm, and 0–10 cm soil layers. Moreover, with increasing soil depth, the number of germinable seeds

showed a downwards trend in both the free-grazing areas and enclosed areas (Figure 4). The statistical analysis of the germination characteristics of the germinable *L. chinensis* seed bank showed that the enclosed area and increasing soil depth and the enclosed areas caused the initial germination time to move backwards (Tables 1 and 2). The results of variance analysis showed that there were significant differences between different soil layers in the enclosed area (Table 2) and between the enclosed area and the free-grazing area in the 0–10 cm soil layer (Table 1) ($p < 0.05$). The germination termination time and germination duration did not show significant differences between the different soil layers in the same treatment or between different treatments in the same soil layer, and there was a lack of unified rules.

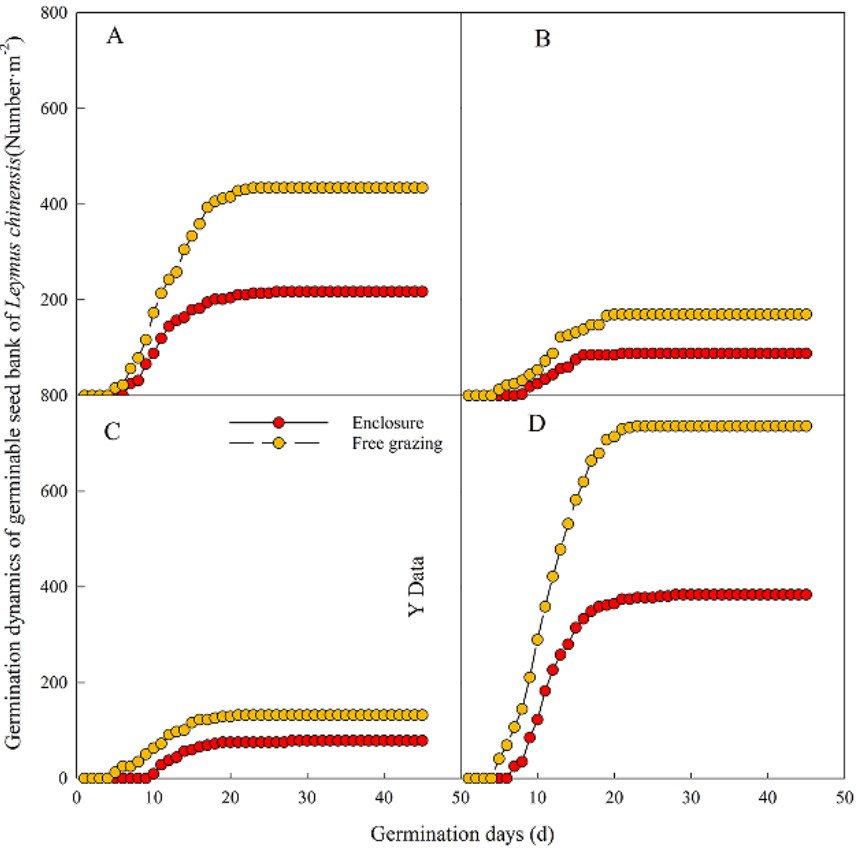

**Figure 4.** Dynamics of *Leymus chinensis* seed germination among the free-grazing and enclosed areas in the same soil layer. Note: Figures (**A**–**D**) show the number of *Leymus chinensis* seed germination at 0–2 cm, 2–5 cm, 5–10 cm, and 0–10 cm, respectively.

**Table 1.** General conditions of the different breeding cattle farms.

| Index | Location | | |
|---|---|---|---|
| | Team 6 | Team 11 | Team 12 |
| Longitude | 120°2′47″–3′36″ E | 120°6′50″–7′28″ E | 120°2′47″–3′36″ E |
| Latitude | 49°19′32″–19′51″ N | 49°20′52″–21′16″ N | 49°19′32″–19′51″ N |
| Annual average temperature | −2.2 °C | −2.4 °C | −2.2 °C |
| Annual precipitation | 351 mm | 350 mm | 351 mm |
| Soil category | chestnut calcareous soil | chestnut calcareous soil | chestnut calcareous soil |
| Main plant populations | *Leymus chinensis, Stipa baicalensis, Artemisia tanacetifolia, Pulsatilla turczaninovii, Cleistogenes squarrosa* | *Leymus chinensis, Stipa baicalensis, Artemisia tanacetifolia, Pulsatilla turczaninovii, Cleistogenes squarrosa* | *Leymus chinensis, Stipa baicalensis, Artemisia tanacetifolia, Pulsatilla turczaninovii, Cleistogenes squarrosa* |
| Dominant species | *Leymus chinensis* | *Stipa baicalensis* and *Leymus chinensis* | *Leymus chinensis* |
| Treatment | Free-grazing area, enclosed area | Free-grazing area, enclosed area | Free-grazing area, enclosed area |

**Table 2.** Effects of the free-grazing and enclosed areas on the *Leymus chinensis* germination seed bank characteristics in the same soil layer.

| Soil Layer | Grassland Use Method | Initial Germination Days | Germination Termination Days | Germination Duration Days | Germination Index |
|---|---|---|---|---|---|
| 0–10 cm | Enclosed area | 8.63 ± 0.42 a | 20.38 ± 1.76 | 12.75 ± 1.92 | 35.44 ± 11.78 |
| | Free-grazing area | 7.00 ± 0.55 b | 19.11 ± 1.02 | 13.11 ± 1.20 | 34.59 ± 6.95 |
| 0–2 cm | Enclosed area | 7.67 ± 0.67 | 22.00 ± 2.65 | 15.33 ± 3.28 | 72.98 ± 22.02 |
| | Free-grazing area | 6.33 ± 0.67 | 22.00 ± 0.58 | 16.67 ± 0.88 | 43.4 ± 18.32 |
| 2–5 cm | Enclosed area | 8.67 ± 0.33 | 17.00 ± 2.08 | 9.33 ± 2.40 | 17.61 ± 4.91 |
| | Free-grazing area | 7.33 ± 1.20 | 18.00 ± 1.53 | 11.67 ± 2.33 | 33.97 ± 8.65 |
| 5–10 cm | Enclosed area | 10.00 ± 0.00 | 23.00 ± 5.00 | 14.00 ± 5.00 | 15.74 ± 9.89 |
| | Free-grazing area | 7.33 ± 1.20 | 17.33 ± 1.86 | 11.00 ± 1.15 | 26.41 ± 9.81 |

Note: Values are given as average ± standard deviation. Lowercase letters represent significant differences between different using methods ($p < 0.05$) according to ANOVA.

The *L. chinensis* seed bank germination index decreased with increasing soil depth. The results of the variance analysis also showed that the *L. chinensis* germination index in the 0–2 cm soil layer was significantly higher than that in the 2–5 cm and 5–10 cm soil layers only in the enclosed area, and the *L. chinensis* germination index in the 0–2 cm soil layer was significantly higher than that in the 2–5 cm and 5–10 cm soil layers in the enclosed area and free-grazing area (Table 2).

The *L. chinensis* seed bank germination index in the 0–2 cm soil layer in the enclosed area was higher than that in the same layer in the free-grazing area, but in the 2–5 cm and 5–10 cm soil layers, the *L. chinensis* seed bank germination index in the enclosed area was lower than that in the free-grazing area. In addition, the *L. chinensis* germination index in the 0–10 cm soil layers was not significantly different between the enclosed and free-grazing areas. The ANOVA results also showed that there was no significant difference in the *L. chinensis* seed bank germination index in this soil layer between the free-grazing and enclosed areas ($p > 0.05$). (Table 3)

**Table 3.** Effects of different soil layers in the free-grazing and enclosed areas on the *Leymus chinensis* seed bank germination characteristics.

| Grassland Use Method | Soil Layer | Initial Germination Days | Germination Termination Days | Germination Duration Days | Germination Index |
|---|---|---|---|---|---|
| Total | 0–2 cm | 7.00 ± 0.52 | 22.00 ± 1.21 | 16.00 ± 1.55 | 58.19 ± 14.42 a |
| | 2–5 cm | 8.00 ± 0.63 | 17.50 ± 1.18 | 10.50 ± 1.59 | 25.79 ± 5.76 b |
| | 5–10 cm | 8.40 ± 0.93 | 19.60 ± 2.34 | 12.20 ± 1.85 | 21.08 ± 6.67 b |
| Enclosed area | 0–2 cm | 7.67 ± 0.67 b | 22.00 ± 2.65 | 15.33 ± 3.28 | 72.98 ± 22.02 a |
| | 2–5 cm | 8.67 ± 0.33 ab | 17.00 ± 2.08 | 9.33 ± 2.40 | 17.61 ± 4.91 b |
| | 5–10 cm | 10.00 ± 0.00 a | 23.00 ± 5.00 | 14.00 ± 5.00 | 15.74 ± 9.89 b |
| Free-grazing area | 0–2 cm | 6.33 ± 0.67 | 22.00 ± 0.58 | 16.67 ± 0.88 | 43.4 ± 18.32 |
| | 2–5 cm | 7.33 ± 1.20 | 18.00 ± 1.53 | 11.67 ± 2.33 | 33.97 ± 8.65 |
| | 5–10 cm | 7.33 ± 1.20 | 17.33 ± 1.86 | 11.00 ± 1.15 | 26.41 ± 9.81 |

Note: Values are given as average ± standard deviation. Lowercase letters represent significant differences between different use methods used ($p < 0.05$) according to ANOVA.

## 4. Discussion

### 4.1. Effect of Grazing on the Germinable Leymus chinensis Seed Bank

*L. chinensis* is widely distributed in East Asia in Eurasia and is also distributed in China, Mongolia, Japan, and eastern Russia [29]. *L. chinensis* is a dominant plant in the natural grassland community in Northeast China, it has high nutritional value and is often used as one of the main plants for vegetation restoration in degraded grassland due to livestock preference in this area [30,31]. However, related study has shown that the seed germination rate of the *L. chinensis* population is often low [14], and the fundamental cause of this phenomenon is that the lemma and endosperm of *L. chinensis* seeds

inhibit their germination [28]. Zhang et al. (2006) [32] studied temperature, light, and different chemical treatments and found that *L. chinensis* seeds have many dormancy mechanisms. Lin et al. (2017) [16] further investigated whether any light, temperature change, or low-temperature treatments were conducive to *L. chinensis* seed germination.

Nevertheless, they found that the germination rate of *L. chinensis* seeds did not exceed 50%. This study compared the *L. chinensis* germination seed bank in enclosed and free-grazing areas and found that the number of germinable *L. chinensis* seeds in the free-grazing areas was significantly higher than that in the enclosed areas. This significant difference was also reflected in the ratio of the number of germinable *L. chinensis* seeds to the total number of germinable seeds in the soil. This result may have occurred because grazing can easily bury *L. chinensis* seeds in the soil, thus improving the germination rate of *L. chinensis*. However, the germination index in Table 1 shows that the *L. chinensis* germination index of the enclosed area in the 0–2 cm soil layer was much greater than that of the free-grazing area, but the *L. chinensis* germination index in the 2–5 cm and 5–10 cm soil layers showed the opposite trend. There was no significant difference in the ANOVA results due to the contrasting rule of the change in the *L. chinensis* germination index in the 0–2 cm, 2–5 cm, and 5–10 cm soil layers. Therefore, whether free-grazing areas could change germination rates needs to be further studied.

Moderate grazing can significantly ensure that the proportion of *L. chinensis* in the community is more than 70% [33]. However, it is difficult to restore the degraded *L. chinensis* grassland plant community to a climax community dominated by *L. chinensis* because the soil seed bank has changed into a decreasing one and there is a lack of source material for restoring the grassland target species [34]. Silva and Overbeck (2020) [35] studied the effect of grazing intensity on grassland seed banks and found that the seed density of soil seed banks increased with increasing grazing intensity, but the composition of dominant species decreased significantly (especially in heavily grazed areas), which means that it will be difficult for the grassland plant community to recover after the disappearance of dominant species. This study indicated that the free-grazing area had little impact on the *L. chinensis* seed bank in the soil of the grassland plant community. This result was consistent with the results of Badgery's research [33], but it promoted the ratio of *L. chinensis* in the soil seed bank, which contradicts the observations of Silva and Overbeck's research [35].

Some studies have noted that heavy grazing can significantly increase the germination rate of *Lespedeza davurica* and can lead to prolonged germination duration [36]. The results of this study were inconclusive due to the germination rate. Nevertheless, Figure 4 shows that the number of germinable *L. chinensis* seeds in the seed bank in the free-grazing area was higher than that in the enclosed area, and there was a tendency for the initial germination time to advance. However, the initial germination time was not significantly earlier in the free-grazing area than that in the enclosed area. Although the advance in this time in the free-grazing area was not significant compared to that in the enclosed area, the cumulative result of this advance trend led to a significant result in terms of the *L. chinensis* seed germination time in the free-grazing area compared with that in the enclosed area.

### 4.2. Effect of Grazing on Seed Source of Germinating Seed Bank of Leymus chinensis

The reproductive strategies of *L. chinensis* in enclosed areas and free-grazing areas were different. An enclosed grassland is conducive to the asexual reproduction of perennial grasses, and more individuals can be ensured through tillering [37,38]. In free-grazing or moderate-grazing areas, the sexual reproduction of *L. chinensis* is dominant because *L. chinensis* can avoid grazing damage through seeds, and study results have shown that although the number of earing branches of *L. chinensis* in the free-grazing area was smaller than that in the enclosed area, the *L. chinensis* seed bearing percentage and seed quality in the free-grazing area were much higher than those in the enclosed area (Qing, 2002) [39]. This study determined the density (number/m$^2$) of the *L. chinensis* population in the free-grazing and enclose grassland areas, showing that there was no significant difference between the *L. chinensis* population density (number/m$^2$) between the two areas (the

density of *L. chinensis* in the free-grazing area was $238.06 \pm 74.85$ number/m$^2$, and the density of *L. chinensis* in the enclosed area was $191.87 \pm 50.34$ number/m$^2$; the data are presented as the mean $\pm$ SE). The results showed that there was no significant difference in the soil germination seed bank (Figure 1), but there was a significant difference in the *L. chinensis* germination seed bank between the free-grazing area and the enclosed area, with that in the free-grazing area being significantly higher than that in the enclosed area (Figure 2). At the same time, the proportion of *L. chinensis* germinating seeds was also higher in the free-grazing area than in the enclosed area (Figure 3). Therefore, combined with the results of previous studies, we believe that in comparison to enclosed areas, free-grazing areas lead to a higher seed setting rate of *L. chinensis* seeds, which leads to a better performing *L. chinensis* germination seed bank in the soil.

This result is consistent with the results of Qing (2002), that is, the number of seedlings in the *L. chinensis* grassland grazing area was much higher than that in the enclosed area, and the number of *L. chinensis* rhizome buds of *L. chinensis* in the enclosed area was significantly higher than that in the grazing area, which is consistent with the previous discussion results, that is, the enclosed area is conducive to the asexual reproduction of *L. chinensis* through the differentiation of rhizome buds into plants, and light grazing and moderate grazing can promote the sexual reproduction of *L. chinensis*.

There will be a trade-off between *L. chinensis* sexual reproduction and asexual reproduction with grazing livestock, which is reflected not only in the seed-bearing percentage but also in the germination rate [37,39,40].

In the free grazing area, due to the influence of grazing livestock, the seeds of *Leymus chinensis* with smaller seeds are more likely to be buried, so there will be a higher germination rate. This scenario is more consistent with the research results of Blumenthal and Ison (1996) [41]. However, in terms of the trade-off between sexual reproduction and asexual reproduction, although Qing (2002)'s research results in 1999 showed that the seed-bearing percentage in the free-grazing area was much higher than that in the enclosed area and that the ratio of healthy seeds to reproductive branches was also higher in the free-grazing area than that in the enclosed area, the number of ears in the enclosed area was lower than that in the free-grazing area.

Thus, while study had shown the relationship between sexual reproduction and asexual reproduction in free-grazing and enclosed areas [39], others had shown that that grazing did not lead to a trade-off between sexual reproduction and asexual reproduction of perennial grasses [42]. In summary, according to the results of this study, it can be inferred that free grazing can improve the germination rate of *L. chinensis* seeds and advance germination time overall. At the same time, because the grazing pressure from free grazing is less than that of moderate grazing, the status and role of *L. chinensis* in the plant community can be well maintained; however, the mechanisms behind this result may need to be studied in combination with seed rain and *L. chinensis* sexual reproduction-related indicators, including the number of reproductive branches, heading number, number of florets per panicle, number of seeds per panicle, and number of germinated seeds, in a further systematic study.

## 5. Conclusions

The moderate free-grazing area did not significantly change the grassland soil *L. chinensis* germinable seed bank. Free-grazing significantly increased the germinating seed number of the *L. chinensis* germinable seed bank and the ratio of the number of *L. chinensis* germinable seeds to the number of total soil germinable seeds in the soil, and overall, the initial germination time of *L. chinensis* in the free-grazing area advanced. With the deepening of the soil layer, the number of germinable seeds and the number of *L. chinensis* germinable seeds showed a downwards trend. Nevertheless, the percentage of *L. chinensis* germinable seeds of the total number of germinable seeds in the soil decreased significantly in the 2–5 cm soil layer. The trampling by livestock in the free-grazing area is beneficial to the burial of *L. chinensis* seeds, thereby increasing their germination rate and seed setting rate through a stress response,

resulting in the densities of *L. chinensis* of the plant communities in the free-grazing area and enclosed area generally remaining consistent.

**Author Contributions:** Conceptualization, H.L., Z.W., X.X. and R.Y.; methodology, H.L. and S.L.; investigation, S.L., B.C. and L.X.; data curation, H.L., S.L., Y.L. and R.Y.; writing—original draft preparation, H.L. and Y.W.; writing—review and editing, Y.L. and R.Y.; funding acquisition, G.Y., R.Y. and X.X. All authors have read and agreed to the published version of the manuscript.

**Funding:** This work was funded by the National Key Research and Development Program of China (2021YFF0703904, 2021YFD1300503), the National Natural Science Foundation of China (31971769, 32130070), the Fundamental Research Funds Central Nonprofit Scientific Institution (1610132021016), the Special Funding for Modern Agricultural Technology Systems from the Chinese Ministry of Agriculture (CARS-34), and the Inner Mongolia Autonomous Region Forestry Research Institute Scientific Research Ability Promotion Project (104004001).

**Data Availability Statement:** Not applicable.

**Acknowledgments:** The authors thank the reviewers and editor for their insightful comments and constructive suggestions.

**Conflicts of Interest:** The authors declare no conflict of interest.

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
