# Peer review of "Free-Grazing versus Enclosure Lead to an Increase in the Germination of the Leymus chinensis Seed Bank in the Hulunbuir Grassland"

_agronomy, doi:10.3390/agronomy13010022_

Round 1
Reviewer 1 Report
I read the manuscript entitled “Free-Grazing Areas versus Enclosed Areas Lead to an Increase in the Germination of the Leymus chinensis Seed Bank in the Hulunbuir Grassland” by Hongmei Liu et al. and I consider that the topic is falling into the scope of the journal and could be of interest for the readers of the journal.
General comment:
The paper is generally well organized, but I recommend to review it in terms of English & writing errors.
Specific comments:
I recommend to consider to change the title as in this current form it looks like a conclusion.
I recommend to review the Abstract section. Generally, this section provides all the information required to be found in this section, but I personally don’t like the formulation. For example: Line 21 “The results were as follows: there was” – this formulation looks like a report.
When using short names it is required to define them at their first appearing in the text. Please see: Line 16 – “L. chinensis”;
Line 36 – “and impacts on the”- please replace with “and impacting ”
Line 40-41 – please introduce a space before reference
Line 53 – “some studies show” please replace with “some studies showed”
Line 55 – “Hopfensperger[20]”- please introduce the missing space before reference. Please check this error throughout the entire text (Lines 62, 65, 71 etc).
Please rephrase the sentence found in Lines 73-77 and highlight that the studies mentioned represent the major objective of your research.
In Materials and Methods the authors mentioned in the first phrase that they covered in their research 3 sampling sites: Site 6, Site 11 and Site 12. But further, in Table 1 these sites are presented as “Team 6, 11 and 12”. Please use the same notation throughout the entire text.
I would recommend to reduce text size or introduce an extra line in the last section of Table 1 because it is difficult to understand the treatments (the text is too tight).
Please rephrase the sentence found on Lines 153-154.
Figure 1 D, 2D, 3 D- please split the two words found in the title – “Soillayer”- correct for “Soil layer”
I would recommend to rephrase the Notes below Figure 1 as this formulation is not scientific and also difficult to understand (for example: what do you mean by “the same below”?)– “Note: There is no significant difference between the same lowercase letters in the figure, P>0.05; the same below.”
Line 252- wrong numeration of Tables from this line to the end of the manuscript – in line 252 should be Table 2 not Table 1.
Please rephrase the Note below Table 1 which should be Table 2 (line 252). Try using a more general explanation like: “Values are given as average ± standard deviation. Effects were accepted as statistically significant if p ≤ 0.05. Values within the same column followed by a common letter are not significantly different according to ANOVA.”
Line 299 – please provide reference for the “Badgery's research”
I would recommend to review the Conclusion section and try to rephrase it having as major points the two questions addressed at the end of Introduction section.
Line 368 – it is not recommended to start your Conclusion with the expression “Compared with ”
Please review the Reference section and list the references in accordance with Agronomy journal’s template.
Reviewer 3 Report
The work is interesting. I read it very carefully anf enjoy that.
Well presented and written.
Meaningful for grazing effect on seed germination in seed banks.
